# Effects of Commercially Available Antioxidant-Enriched Fish- and Chicken-Based Diets on Biochemical Parameters and Blood Fatty Acid Profile of Old Dogs

**DOI:** 10.3390/ani12101326

**Published:** 2022-05-23

**Authors:** Sara Sechi, Silvia Carta, Fabio Correddu, Alessandro Di Cerbo, Anna Nudda, Raffaella Cocco

**Affiliations:** 1Department of Veterinary Medicine, Pathology and Veterinary Clinic Section, University of Sassari, Via Vienna 2, 07100 Sassari, Italy; sarasechilavoro@tiscali.it; 2Department of Agriculture, University of Sassari, Via de Nicola 9, 07100 Sassari, Italy; scarta2@uniss.it (S.C.); fcorreddu@uniss.it (F.C.); anudda@uniss.it (A.N.); 3School of Biosciences and Veterinary Medicine, University of Camerino, 62024 Matelica, Italy; alessandro.dicerbo@unicam.it

**Keywords:** blood fatty acids, dog diet, antioxidant, biochemical parameters

## Abstract

**Simple Summary:**

The effects of different diets on the biochemical parameters and fatty acid profile of dogs’ blood were investigated. The dog diet is normally composed of chicken or fish meat, which represents a good source of protein and fat. However, fish and chicken meat contain a high amount of lipids susceptible to oxidation, which could alter the beneficial effects of the dog food. The addition of antioxidant compounds to the diet might improve the quality of the diet and, consequently, the health of the animals.

**Abstract:**

The aim of this study was to evaluate the effects of two commercially available fish- (FH) and chicken (CH)-based diets and the same diets combined with antioxidant compounds, e.g., FH-AOX and CH-AOX, respectively, on biochemical parameters and blood fatty acid (FA) profile. For this purpose, 28 dogs were allocated to four groups and fed the four diets for six months. Blood samples were taken before starting the experimental period and at the end of the trial. Concerning the biochemical parameters, the animals fed CH-AOX reached the highest values for ALB compared to the animals fed CH or FH-AOX; however, the values were within the reference values for old dogs. Triglycerides and urea were significantly higher in FH compared to the CH diet; however, both FH-AOX and CH-AOX showed a marked decrease in the TRI and urea concentration with respect to FH and CH. Moreover, CH-AOX and FH-AOX increased glucose values, linoleic acid content, and polyunsaturated FA n-3 with respect to FH and CH. On the contrary, arachidonic acid was lower in the CH-AOX and FH-AOX groups. In conclusion, FH led to a better FA profile than that of the CH diet, while CH-AOX and FH-AOX improved the FA profile regardless of the basal diet.

## 1. Introduction

Blood fatty acid composition could be used as a biomarker of dietary intake and could be an important parameter to understand the relationship between diet, metabolism, and some diseases. Some fatty acids (FA), such as polyunsaturated fatty acids (PUFA), have a potential role in the prevention of several diseases, such as atherosclerosis, cancer, and inflammation [1,2,3,4,5,6,7,8]. A positive correlation between the total content of dietary PUFA in the diet and the relative amount of PUFA in the blood lipid fraction has been documented in humans [9,10] and animals [11].

There is scant literature about the effect of the lipid composition of the diet on the blood biochemical parameters and fatty acid profile in dogs. Some studies showed that the incorporation of PUFA-3 into dog nutrition could be beneficial in early renal insufficiency [12], in the prevention of atrial fibrillation [13], and in atopic dermatitis [14,15]. 

Stoeckel et al. showed that the incorporation of n-3 PUFA into the dog diet leads to an increase in docosahexaenoic acid (DHA) and eicosapentaenoic acid (EPA) in erythrocytes in a dose-dependent manner [14].

Fish and chicken are two of the main meat ingredients used in dry dog food [16,17,18,19,20,21]. However, the chemical composition of these two ingredients is different and could influence the health status of the animals in different ways. Fish and chicken have different lipid compositions; in fact, fish are richer in long-chain PUFA (LC-PUFA) n-3, mostly EPA, docosapentaenoic acid (DPA), and DHA, with respect to chicken. However, PUFA are more susceptible to oxidation, compared to saturated (SFA) and monounsaturated fatty acids (MUFA), a process that happens frequently during food storage, processing, cooking, and even during gastrointestinal digestion, increasing the amount of quite reactive aldehydic end-products, particularly 4-hydroxyhexenal [22]. For this reason, the synergic action of antioxidant compounds and PUFA in the diet could be useful to ameliorate the antioxidant stability of foods and the health status of the animals.

The aim of this research was to evaluate the effects of four different diets, containing fish and chicken meat alone or combined with natural antioxidant compounds, on hematological and biochemical parameters as well as the blood fatty acid profile of dogs.

## 2. Materials and Methods

### 2.1. Experimental Design, Dogs, and Diets

This evaluation was designed as a randomized one. Twenty-eight, healthy mixed breed dogs (mean age ± SD; 9.09 ± 2.79 years and mean weight ± SD; 22.01 ± 1.17 kg; 57.1% males, 42.9% females) with a Body Condition Score ranging from 15 to 30% were randomly allocated into 4 groups, having both sexes in each group.

Only 3 out of 12 females (mean age 13 years) presented a heart murmur at auscultation, and the echography showed the presence of a compensated endocardiosis of the mitral valve. All dogs lived in a shelter placed in Andriolu Porto Torres (Sassari, Sardinia, Italy) within 8 m^2^ covered boxes with an outer fence of 15 m^2^, which hosted 2 dogs each. All dogs were allowed to go out twice a day and they spent this time smelling the environment and going for short walks along with shelter operators. Twice a day (8.00 a.m. and 7.00 p.m.), a veterinarian was responsible for the diet provision according to the manufacturer’s instructions (1–10 kg → 30–180 g; 11–20 kg → 190–300 g; 21–35 kg → 310–455 g). No adverse effects were observed for each dog following the diet administration. All animals were provided water ad libitum. The overall trial lasted 6 months.

The first group was fed a commercially available chicken-based organic diet (group CH), the second with a commercially available chicken-based organic diet + natural antioxidant compounds (group CH-AOX), the third group was fed a commercially available fish-based diet (group FH), and the fourth group a commercially available fish-based organic diet + natural antioxidant compounds (group FH-AOX). The daily amount of diet was suggested by the manufacturer.

### 2.2. Sample Collection and Analysis

Blood samples were taken from the cephalic vein of each dog at the beginning (T_0_) and at the end of the trial (T_1_) and split into two aliquots, one rapidly processed for biochemical analysis and the other (with heparin) stored at −80 °C for fatty acid analysis.

### 2.3. Pet Food Sample Analysis

Pet food samples were analyzed for dry matter (DM) (105 °C for 24 h), crude protein (CP), ether extract, ashes [23], neutral detergent fiber (NDF) (using heat-stable α-amylase) [24], acid detergent fiber (ADF) [25], and acid detergent lignin (ADL) [26]. All parameters were expressed as a percentage of DM. The fatty acid content in pet food samples was analyzed as detailed by Correddu et al. [27].

The antioxidant compounds used in CH-AOX and FH-AOX are detailed in Table 1.

All diets fulfilled the recommendations for dietary ingredients regarding dog daily requirements [16,28]. Briefly, the nutrient composition was 24% of crude protein, 12% of crude oils and fats, 3.7% of crude fiber, 5% of crude ash, 9% of moisture, and a metabolized energy of 3.477 kcal/kg (or 14.6 MJ/kg). Both diets were in the form of kibble industrially produced and had the same amount of vitamins (A, C, and E), trace elements (choline chloride, zinc sulfate monohydrate, and cupric chelate glycine hydrate), and amino acids (DL-methionine).

### 2.4. Biochemical Analysis

Blood samples for biochemical analysis were centrifuged at 4000× *g* 1.5 min at 37 °C and serum was used to assess albumin (ALB), alkaline phosphatase (ALP), total bilirubin (BT), calcium (CAL), cholesterol (COL), creatine kinase (CK), creatinine (CRE), gamma-glutamyl transferase (GGT), glucose (GLU), serum glutamic oxaloacetic transaminase (GOT), serum glutamic-pyruvic transaminase (GPT), phosphorous (P), total protein (PRO), triglycerides (TRI), and urea (UR) using an automatic chemistry analyzer (Dimension RXL, Siemens S.p.A, Milan, Italy).

### 2.5. Blood Fatty Acid Profile

Whole blood samples of T_1_ were analyzed for fatty acid methyl esters (FAME). The lipid extraction was performed according to the method proposed by Folch et al. [29] with some modifications. Briefly, the lipids were extracted from 0.5 mL of whole blood using a 2:1 chloroform–methanol solution, and then the solution was homogenated and filtrated. A solution of NaCl (1 M) was added to the tubes and, after the centrifugation, the aqueous phase was removed, and the chloroform–methanol solution was dried up with nitrogen.

The FA profile was determined using the method of Kramer et al. [30] with some modifications. Briefly, 1 mL of sodium methoxide 0.5 M in methanol (Sigma and Aldrich, Spain) was added to the extracted lipids and then the samples were placed in a water bath at 50 °C for 10 min. After cooling to room temperature, a solution of HCl/methanol (3 M, 1.5 mL for each sample) was added and the samples were placed in the water bath at the previous conditions. After this, 1 mL of a solution containing methyl nonadecanoate (C19:0) as internal standard (Sigma Chemical Co., St. Louis, MO, USA) and 3.5 mL of a 0.43 M solution of K_2_CO_3_ were added. The samples were centrifuged (1500× *g*, room temperature, 5 min) and the supernatant was withdrawn and stored in vials for gas chromatography (GC) analysis.

The FA were analyzed with a 7890A GC System (Agilent Technologies, Santa Clara, CA, USA), equipped with a 7693 Autosampler (Agilent Technologies, Santa Clara, CA, USA) and a flame ionization detector (FID). Helium was the carrier gas (1 mL/min flow rate). The temperature of the instrument had an increase of 13 °C/min, from 45 °C to 175 °C, and an increase of 4°C/min until 215 °C. The injector and the detector had a temperature of 250 °C. The area of the FAME was determined using the OpenLAB CDS GC ChemStation Upgrade software data system (Revision C.01.04, Agilent Technologies Inc., Santa Clara, CA, USA) and the peaks were identified by comparing their retention time with those of methyl standard.

The FA were reported as g/100 g of total FAME and groups of FA were calculated as follows: SFA—sum of the individual saturated fatty acids; UFA—sum of the individual unsaturated fatty acids; MUFA—sum of the individual monounsaturated fatty acids; PUFA—sum of the individual polyunsaturated fatty acids; OBCFA—sum of individual odd- and branched-chain fatty acids; short-chain fatty acids (SCFA)—sum of the individual fatty acids from C4:0 to C10:0; medium-chain fatty acids (MCFA)—sum of the individual fatty acids from C11:0 to C17:0; long-chain fatty acids (LCFA)—sum of the individual fatty acids from C18:0 to C22:6 (DHA); PUFA n-3—sum of individual n-3 fatty acids; PUFA n-6—sum of individual n-6 fatty acids; total CLA—sum of individual conjugated linoleic acids. The nutritional properties of blood fat were estimated by the n-6 to n-3 ratio and three indices, such as the atherogenic index (AI) and trombogenic index (TI), were calculated according to Ulbricht and Southgate [31], except for the substitution of C18:0 with C12:0, as suggested by Nudda et al. [32]:AI = [12:0 + (4 × 14:0) + 16:0]/[(PUFA) + (MUFA)],
TI = (14:0 + 16:0)/[(0.5 × MUFA) + (0.5 × n-6) + (3 × n-3) + (n-3:n-6)];

The hypocholesterolemic to hypercholesterolemic ratio (h:H) was calculated according to Fernández et al. [33]: h:H = [(sum of 18:1cis-9, 18:1cis-11, 18:2 n-6, 18:3 n-6,18:3 n-3, 20:3 n-6, 20:4 n-6, 20:5 n-3, 22:4 n-6, 22:5 n-3 and 22:6 n-3)/(14:0 + 16:0)].

### 2.6. Statistical Analysis

All data (biochemical and FAME) were analyzed using the two-way ANOVA GLM procedure of SAS version 9.2 (SAS Institute, Cary, NC, USA, 2008), in which the diet, AOX supplementation, and their interaction were used as fixed effects. A *p* < 0.05 was considered as significant.

## 3. Results and Discussion

The chemical compositions of the CH, FH, and AOX are reported in Table 2. The FH showed similar protein and fat content compared to CH. The lipid composition of FH was characterized by a high proportion of alpha-linolenic acid (18:3 c9, c12, c15; n-3) and LC-PUFAn-3 such as EPA, DPA, and DHA. In FH and CH, oleic acid (18:1 c9; n-9), palmitic acid (16:0), and linoleic acid (18:2 c9, c12; n-6 LA) were in a similar proportion. The supplement AOX, composed of vegetables, had low fat content (0.5%), in which LA was the main FA.

### 3.1. Biochemical Parameters

Biochemical parameters are shown in Table 3.

Almost all of the biochemical parameters were within the optimal range for old dogs. The diet did not influence any parameters, while the supplement influenced some of them, such as GLU and P, which reached the highest values in animals belonging to the AOX group.

TRI and UR were influenced by the interaction D*S; in particular, TRI was higher in animals fed FH and that did not receive supplementation compared to the other groups.

The depressive effect of polyphenols on blood TRI has been reported in laboratory animals [34] and humans [35].

A similar trend was found for UR, which achieved the highest values in animals fed FH and that did not receive the AOX supplementation. Similar results were observed in broilers fed fish meal or poultry by-product meal [36]. The authors ascribed this effect to the lower protein biological value of fish meal compared to the other diet. The AOX supplementation decreased the urea concentration, significantly, in the FH diet and, numerically, in the CH diet. This result could be related to the polyphenol content of the AOX supplement, which can lead to improved nitrogen metabolism and reduced urea production. This finding was in accordance with previous work on pigs fed a polyphenol supplement [37].

### 3.2. Blood Fatty Acid Profile

The effect of the diet, the AOX supplementation, and their interaction on the blood fatty acid profile is shown in Table 4.

The diets influenced several FA, including some minor FA, such as C16:1 trans-6-7, anteisoC17:0, and C16:1c9. The C18:0 was influenced by the diet and the supplement. As regards the influence of the diet, the highest value was found in the CH group, while, concerning the effect of the supplement, it was found in the AOX group. This is partly due to the highest content of C18:0 in chicken and the presence of this FA in the antioxidant supplement. In contrast, elaidic acid (C18:1 trans-9) was influenced by the supplement, but not by the diet, and the highest value was achieved by dogs that did not receive any supplementation. In the same manner, linoleic acid (C18:2n6) was influenced by the supplement and the highest values were found in animals fed CH-AOX and FH-AOX. This effect could arise from three aspects: (i) the amount of this FA in the AOX, (ii) the reduction in the linoleic oxidation extent related to the antioxidant effect of the supplement [38], (iii) the inhibition of elongase and desaturase activity. This last hypothesis seems to be confirmed by the lower values of arachidonic acid (ARA, C20:4n6) found in AOX groups compared to N-AOX groups. Some studies demonstrated that polyphenols could decrease the activity of Δ6-desaturase and also could reduce the gene expression for Δ9-desaturase [39,40], enzymes involved in LC-PUFA synthesis. This is confirmed also by a study carried out by Kojadinovic et al. [41] in which pomegranate juice rich in polyphenols and antioxidant compounds was administrated to women with metabolic syndrome and the main consequence was the reduction in the ARA in the blood fatty acid profile. Linolenic acid (C18:3n3) was higher in the FH than CH diet, according to the higher concentration of this FA in the FH diet.

The inclusion of the AOX supplementation did not affect the content of this FA. The diet and the supplement influenced the content of EPA and DHA, which were higher in FH than CH, but it was markedly increased by the addition of an antioxidant to the fish diet.

In vivo and in vitro studies showed that polyphenols exerted a positive effect on the prevention of peroxidation of oils, leading to an improvement in the absorption of LC-PUFA3 in the intestine [42,43]. Thus, we could hypothesize a direct positive effect of the polyphenols contained in the AOX supplement on the uptake of these FA in the dog intestine. In addition, contrary to the consideration of the reduction in ARA in AOX diets, the increase in some LC-PUFA n-3 seems to be related to an increment in the conversion rate of linolenic acid (C18:3n3) to their longer derivatives, due to the presence of polyphenols [44].

This is in agreement with other studies on humans supplemented with flavanol-rich cocoa and other polyphenols [44,45]. A study conducted on rats showed that a diet rich in polyphenols increased the content of EPA and DHA in blood, suggesting that these compounds could alter the activity of the desaturation enzymes [44]. Our study suggested that antioxidant compounds interfere with PUFA n-3 metabolism and lead to an accumulation of these FA, independently from other fat supplies. Moreover, a recent review confirmed that polyphenols can regulate LC-PUFA synthesis, by acting on the desaturation pathway [46].

However, in our study, the decrease in LNA, observed in the FH-AOX, is too low to explain in total the large increase in EPA in the same groups.

Instead, several studies in animals and humans showed that a low ARA/EPA ratio may be considered a marker of cardioprotective effects, such as anti-inflammatory effects, inhibition of platelet aggregation, or effects on the vascular endothelium [47].

Moreover, the PUFA n-3 arising from fish could have beneficial effects on some clinical disorders in dogs, such as dermatologic, cardiovascular, renal, and metabolic diseases [48]. Although the effect of fish fat on animal health is known, the effect of polyphenols and the antioxidant compound is still unclear. However, our study seems to demonstrate that polyphenols could exert a positive effect on blood fat, independently from the basal diet administrated. More studies are necessary to better understand the pathway by which polyphenols impact the FA profile and, consequently, animal health.

As regards FA groups, OCFA and OBCFA were affected by the supplement, achieving lower values in animals belonging to the AOX group. The diet and the supplement affected PUFA 3 and the ratio n3:n6. These groups of FA were higher in the FH and AOX groups, as a demonstration of the highest quality of fish fat profile and the positive effect of the antioxidant on dogs’ health.

As regards the quality of the blood FA profile, although the FH diet was richer in beneficial FA than the CH one, no difference was observed in the AI, TI, and h:H indices, regardless of the use of AOX.

## 4. Conclusions

In conclusion, commercially available fish- and chicken-based diets and the antioxidant supplement affected some biochemical parameters and the fatty acid profile of dogs’ blood. TRI and UR were higher in animals fed FH and that did not receive supplementation than dogs belonging to the other groups. The blood fatty acid profile reflected the different fatty acid compositions of the fish and chicken meals. Dogs fed the fish diet had higher PUFA, such as linolenic acid, EPA, and DHA, than the dogs fed the chicken diet. Moreover, the inclusion of antioxidant compounds in the two diets had a positive effect on blood fatty acids, as demonstrated by the increase in PUFA n-3.

## Figures and Tables

**Table 1 animals-12-01326-t001:** Average substances administered to dogs depending on body weight (considering medium body weight).

Antioxidant Substances	Amount Per kg of Complete Food
*Grifola frondosa*	270 mg/kg
*Curcuma longa*	102 mg/kg
*Carica papaya*	135 mg/kg
*Punica granatum*	70 mg/kg
*Aloe vera*	135 mg/kg
*Polygonum cuspidatum*	7 mg/kg
*Solanum lycopersicum*	250 mg/kg
*Vitis vinifera*	24 mg/kg
*Rosmarinus officinalis*	15.6 mg/kg

**Table 2 animals-12-01326-t002:** Ingredients and chemical composition and fatty acid (FA) profile of two diets and the antioxidant supplement.

	Diet ^1^		AOX
Item ^2^	FH	CH	
Chemical composition (% of DM unless otherwise noted)			
DM (%)	91.64	92.11	91.00
NDF	16.17	14.54	7.01
ADF	7.24	5.49	-
ADL	2.92	1.85	-
CP	23.92	24.23	10
Ash	8.23	8.69	6.03
Crude fat	11.00	12.00	0.50
Major FA (g/100 g of total FA)			
C14:0	1.00	0.86	-
C16:0	18.33	19.91	22.56
C16:1c9	2.89	2.91	-
C18:0	4.78	6.38	4.26
C18:1c9	35.87	36.33	11.18
C18:1c11	2.35	2.10	0.00
C18:2n6	29.10	28.73	55.57
C18:3n3	2.76	1.86	6.43
C20:4n6	0.57	0.56	Nd
C22:5 n-3 (EPA)	0.67	0.08	Nd
C22:5 n-3 (DPA)	0.26	0.08	Nd
C22:6 n-3 (DHA)	1.41	0.19	Nd
Metabolizable energy	14.3	14.5	Nd

^1^ Diet: FX = commercial dry food with fish, ingredients: fish 21%, corn, rice bran, beet pulp, poultry fat, corn gluten, animal products, fish oil, minerals, oligosaccharides, FOS, Yucca schidigera; additives (per kg): vitamin A 17000 UI, vitamin D3 1000 UI, vitamin E/all-rac-alpha-tocopheryl acetate 240 mg, choline chloride 1400 mg, zinc sulfate monohydrate 137 mg (Zn 50 mg), Copper(II) chelate of amino acids hydrate 39 mg (Cu 10 mg), DL Methionine 500 mg. CH = commercial dry food with organic chicken, ingredients: corn, sorghum, poultry protein 14%, fish, potatoes 5%, beet pulp, poultry fat, brewer’s yeast, fish oil, minerals, dehydrated yeast (Bio MOS), FOS, Yucca schidigera; additives (per kg): vitamin A 15500 UI, vitamin D3 1000 UI, vitamin E/all-rac-alpha-tocopheryl acetate 240 mg, choline chloride 1000 mg, zinc sulfate monohydrate 137 mg (Zn 50 mg), Copper(II) chelate of glycine hydrate 39 mg (Cu 10 mg), DL Methionine 500 mg, Omega 3/6 ratio of 1:0.8. AOX = antioxidant supplement (a mixture of *Grifola frondosa, Curcuma longa, Carica papaya, Punica granatum, Aloe vera, Polygonum cuspidatum, Solanum lycopersicum, Vitis vinifera,* and *Rosmarinus officinalis*); ^2^ EPA = eicosapentaenoic acid; DPA = docosapentaenoic acid; DHA = docosahexaenoic acid. Nd = not detected.

**Table 3 animals-12-01326-t003:** Effect of different diets on biochemical parameters in dogs.

Item ^1^	Reference Values	Diet ^2^		Supplement		SEM ^3^	*p*-Value		
		FH	CH	AOX	N-AOX		Diet	Supplement	D*S
ALB, g/dL	2–3.3	2.76	2.83	2.87	2.71	0.04	0.348	0.071	0.183
ALP, U/L	1.5–90	83.64	95.94	111.00	61.73	15.21	0.500	0.124	0.834
BT, mg/dL	0.05–0.5	0.08	0.07	0.08	0.07	0.01	0.672	0.401	0.832
CAL, mg/dL	8–10	10.70	10.71	10.73	10.68	0.07	0.956	0.912	0.477
COL, mg/dL	80–250	254.91	230.19	234.19	249.09	11.78	0.177	0.289	0.399
CPK, U/L	100–250	125.73	175.88	147.81	166.55	14.21	0.145	0.765	0.857
CRE, mg/dL	0.5–1.5	1.36	1.29	1.34	1.29	0.03	0.273	0.580	0.890
GGT, U/L	6–16	8.73	8.50	8.88	8.18	0.57	1.000	0.574	0.851
GLU, mg/dL	50–100	65.73	56.56	69.13	47.45	2.82	0.299	<0.0001	0.943
GOT, U/L	25–72	41.55	43.50	41.69	44.18	1.42	0.788	0.401	0.540
GPT, U/L	30–85	75.55	63.38	80.69	50.36	15.84	0.984	0.357	0.654
P, mg/dL	3.5–6.5	3.45	3.51	3.68	3.19	0.11	0.416	0.022	0.970
PRO, g/dL	5.3–8.3	7.28	7.02	7.28	6.91	0.09	0.179	0.153	0.280
TRI, mg/dL	23–100	81.00	68.81	66.00	85.09	9.44	0.124	0.072	0.030
UR, mg/dL	20–50	38.09	32.94	31.75	39.82	1.71	0.003	0.000	0.022

^1^ ALB = albumin blood test; ALP = alkaline phosphatase; BT = total bilirubin; CAL = calcium; COL = cholesterol; CRE = creatinine; GGT = gamma-glutamyl transferase; GLU = glucose; GOT = serum glutamic oxaloacetic transaminase; GPT = serum glutamic-pyruvic transaminase; P = phosphorus; PRO = total protein; TRI = triglycerides; UR = urea; ^2^ FH = diet with fish; CH = diet with organic chicken; AOX = antioxidant supplement; N-AOX = without antioxidant; D*S = diet and antioxidant supplement. ^3^ SEM = standard error of the mean.

**Table 4 animals-12-01326-t004:** Effect of different diets on fatty acid profile and nutritional indices of whole blood.

FAME (g/100 g of Total FA) ^1^	Diet ^2^		Supplement		SEM ^3^	*p*-Value		
	FH	CH	AOX	N-AOX		Diet	Supplement	D*S
C14:0	0.387	0.391	0.377	0.408	0.015	0.980	0.411	0.868
C15:0	0.156	0.145	0.143	0.159	0.006	0.285	0.169	0.998
C16:0	18.285	18.474	18.083	18.855	0.370	0.976	0.378	0.964
C16:1 trans-6-7	0.085	0.041	0.071	0.041	0.005	<0.0001	0.001	0.355
isoC17:0	0.120	0.113	0.110	0.123	0.011	0.962	0.791	0.220
C16:1c7	0.386	0.402	0.378	0.421	0.022	0.687	0.592	0.348
anteisoC17:0	0.048	0.053	0.039	0.069	0.004	0.906	<0.0001	0.114
C16:1c9	0.665	0.783	0.664	0.838	0.035	0.483	0.005	0.056
C17:0	0.618	0.590	0.597	0.607	0.015	0.522	0.853	0.290
isoC18:0	0.102	0.113	0.103	0.118	0.014	0.615	0.883	0.404
C17:1 c6-7	0.027	0.028	0.026	0.030	0.002	0.555	0.791	0.051
C17:1c8	0.228	0.221	0.232	0.213	0.007	0.941	0.161	0.406
C17:1c9	0.051	0.061	0.049	0.070	0.008	0.423	0.469	0.125
C18:0	25.569	26.684	26.784	25.423	0.314	0.012	0.005	0.677
C18:1t9	0.184	0.189	0.155	0.233	0.011	0.420	<0.0001	0.788
C18:1c9	11.246	11.708	10.960	12.334	0.292	0.786	0.008	0.072
C18:1c11	2.953	3.108	3.016	3.088	0.065	0.196	0.970	0.304
C18:2n6 (LA)	22.098	20.448	22.636	18.915	0.522	0.186	0.001	0.248
C18:2t11c15	0.254	0.223	0.232	0.240	0.006	0.009	0.180	0.673
C20:0	0.152	0.132	0.153	0.121	0.008	0.226	0.180	0.178
C18:3n6	0.068	0.072	0.063	0.081	0.005	0.664	0.259	0.100
C20:1c11	0.598	0.433	0.555	0.420	0.027	0.009	0.014	0.259
C18:3n3 (LNA)	0.138	0.093	0.109	0.116	0.008	0.000	0.069	0.065
C20:2n6	0.483	0.377	0.474	0.342	0.020	0.025	0.001	0.368
C20:3n9	0.023	0.073	0.028	0.088	0.013	0.080	0.120	0.250
C22:0	0.116	0.104	0.111	0.107	0.006	0.398	0.918	0.955
C20:3n6	0.999	0.991	1.062	0.896	0.054	0.498	0.066	0.170
C22:1n9	0.079	0.065	0.073	0.068	0.003	0.033	0.798	0.834
C20:4n6 (ARA)	10.368	11.261	9.540	12.871	0.500	0.639	0.002	0.303
C22:5 n-3 (EPA)	1.020	0.375	0.907	0.245	0.095	0.000	<0.0001	0.076
C24:0	0.204	0.169	0.190	0.174	0.010	0.043	0.915	0.085
C22:3n6	0.144	0.134	0.147	0.126	0.007	0.473	0.297	0.229
C25:0	0.327	0.534	0.274	0.704	0.051	0.011	<0.0001	0.020
C22:5 n-3 (DPA)	0.665	0.555	0.568	0.645	0.037	0.138	0.250	0.497
C22:6 n-3 (DHA)	1.076	0.794	1.016	0.752	0.057	0.046	0.048	0.732
MCFA	21.160	21.417	20.872	21.951	0.409	0.959	0.269	0.896
LCFA	78.840	78.583	79.128	78.049	0.409	0.959	0.269	0.896
SFA	46.084	47.504	46.965	46.868	0.497	0.146	0.603	0.673
MUFA	16.504	17.040	16.179	17.757	0.329	0.909	0.011	0.185
PUFA	37.412	35.457	36.857	35.375	0.649	0.277	0.401	0.771
UFA	53.916	52.496	53.035	53.132	0.497	0.146	0.603	0.673
OCFA	1.101	1.269	1.015	1.470	0.054	0.062	<0.0001	0.009
BCFA	0.270	0.280	0.252	0.310	0.027	0.790	0.518	0.271
OBCFA	1.371	1.548	1.267	1.780	0.069	0.177	0.000	0.021
PUFA6	34.161	33.283	33.923	33.231	0.562	0.578	0.671	0.928
PUFA3	2.898	1.817	2.600	1.758	0.160	0.002	0.008	0.286
n6_n3	12.470	19.417	14.196	20.064	1.006	0.000	0.004	0.961
n3_n6	0.084	0.054	0.076	0.052	0.004	0.001	0.003	0.300
CLA	0.076	0.061	0.073	0.059	0.004	0.212	0.036	0.070
AI	0.372	0.385	0.373	0.389	0.011	0.625	0.671	0.730
TI	0.555	0.620	0.566	0.634	0.021	0.253	0.253	0.893
h:H	2.767	2.647	2.746	2.624	0.076	0.599	0.547	0.913

^1^ FAME = fatty acid methyl ester; SA = stearic acid; LA = linoleic acid; LNA = linolenic acid; ARA = arachidonic acid; EPA = eicosapentaenoic acid; DPA = docosapentaenoic acid; DHA = docosahexaenoic acid; SFA = sum of the individual saturated fatty acids; UFA = sum of the individual unsaturated fatty acids; MUFA = sum of the individual monounsaturated fatty acids; PUFA = sum of the individual polyunsaturated fatty acids; OCFA = odd-chain fatty acids; BCFA = branched-chain fatty acids, sum of iso- and anteiso-FA; OBCFA = odd- and branched-chain fatty acids, sum of odd-, iso-, and anteiso-FA; MCFA = medium-chain fatty acids, sum of the individual fatty acids from C11:0 to C17:0; LCFA = long-chain fatty acids, sum of the individual fatty acids from C18:0 to DHA; PUFA n-3 and PUFA n-6 = sum of individual n-3 and n-6 fatty acids, respectively; CLA = sum of individual conjugated linoleic acids; TI = thrombogenic index; AI = atherogenic index; h:H = hypocholesterolemic to hypercholesterolemic ratio; ^2^ FH = diet with fish; CH = diet with organic chicken; AOX = antioxidant supplement; N-AOX = without antioxidant; D*S = diet and antioxidant supplement. ^3^ SEM = standard error of the mean.

## Data Availability

The data presented in this study are available on request from the corresponding author.

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
