# Peer review of "Effects of Commercially Available Antioxidant-Enriched Fish- and Chicken-Based Diets on Biochemical Parameters and Blood Fatty Acid Profile of Old Dogs"

_animals, 2022, doi:10.3390/ani12101326_

Round 1

Reviewer 1 Report

The study is interesting and adds to the literature. However, it needs a lot more discussion and a major English revision. I was correcting some of it, but gave up after line 47. The science is sound, but to be considered for publication the authors need to do a lot of work on this article.

L 11- composed of

L12- substitute lipid by fat

L17- ..with antioxidants supplementation (FH-AOX……

L18- remove comma before “on biochemical”

L23- “In regard to..”

L26- remove “found”

L27- than the CH diet.

L27- Is it in addition or in conclusion?

L27- “..supplement improved the FA profile..”

L35- ..diseases such as…

L36- content of dietary PUFA

L45- “..used in dry dog food”

L46- “could influence the health status of the animals in different ways.

L47- Have different lipid composition

Materials and Methods- How many dogs allocated per group? If you had 4 treatments, shouldn’t you have 28 dogs instead of 27?

L 66- What is a box? Do you mean kennel? It needs dimensions.

What was the experimental design? It seems that it was a complete randomized design, but it is not written in the M&M.

L74- Pet food is referred as “food” and not “feed”.

L80- There is no daily requirement for carbohydrates. Carbohydrates are not essential.

L82-83- were these percentages measured or calculated? And which diet are you referring to? They can’t be the same.

L89- was serum analyzed in sequence after centrifugation, or did you freeze it first?

L140- "..using the general linear model (GLM) procedure..” Did you not have any random effects? I suppose only diet as fixed effect would be enough for a CRD.

Table 1- It can’t have the two diets and the supplement under the title “Diet”. What is the dietary composition of the actual diets? I can’t find it on the Tables.

Table 1- where are the ingredients (according to the Table title)?

Table 2- remove “T0 values were included as a 160 covariate. .”

Table 3- how did you obtain a SEM if the number of dogs in each treatment was not the same? If you had 27 dogs, you couldn’t have had 7 dogs per treatment. SEM is only the same for all diets if you have the same number of replicates.

L243- chicken meal?

It needs more discussion. What is the effect these dietary changes might have on the animals? You need to consider that values were within the reference range, and that some changes, although statistically significant, do not mean they were biologically significant. Also, a change of 0.06 vs 0.04 g/100g is not meaningful, it might as well be analytical variance.

Author Response

The study is interesting and adds to the literature. However, it needs a lot more discussion and a major English revision. I was correcting some of it, but gave up after line 47. The science is sound, but to be considered for publication the authors need to do a lot of work on this article.

Authors: the authors thank the reviewer for the useful suggestions and comments.

L 11- composed of

Authors: we corrected it.

L12- substitute lipid by fat

Authors: we corrected it.

L17- ..with antioxidants supplementation (FH-AOX……

Authors: we corrected it.

L18- remove comma before “on biochemical”

Authors: we corrected it.

L23- “In regard to..”

Authors: we corrected it.

L26- remove “found”

Authors: we amended the sentence.

L27- than the CH diet.

Authors: we corrected it.

L27- Is it in addition or in conclusion?

Authors: we corrected it.

L27- “..supplement improved the FA profile..”

Authors: we corrected it.

L35- ..diseases such as…

Authors: we corrected it.

L36- content of dietary PUFA

Authors: we corrected it.

L45- “..used in dry dog food”

Authors: we corrected it.

L46- “could influence the health status of the animals in different ways.

Authors: we corrected it.

L47- Have different lipid composition

Authors: we corrected it.

Materials and Methods- How many dogs allocated per group? If you had 4 treatments, shouldn’t you have 28 dogs instead of 27?

Authors: we apologize for the mistake; the number of enrolled dogs was 28.

L 66- What is a box? Do you mean kennel? It needs dimensions.

Authors: we apologize for the lack of information. We provided them in the manuscript

What was the experimental design? It seems that it was a complete randomized design, but it is not written in the M&M.

Authors: we reported the information in the M&M section. We modified the statistical analysis (and the related results) in accordance to the request of the second reviewer.

L74- Pet food is referred as “food” and not “feed”.

Authors: we corrected it.

L80- There is no daily requirement for carbohydrates. Carbohydrates are not essential.

Authors: thanks for the comment. We corrected the sentence

L82-83- were these percentages measured or calculated? And which diet are you referring to? They can’t be the same.

Authors: we agree with the reviewer. Actually, the chemical composition of the 2 diets is reported in the table 1. We corrected the sentence and added this information in the text.

L89- was serum analyzed in sequence after centrifugation, or did you freeze it first?

Authors: the serum was analyzed immediately without freezing.

L140- "..using the general linear model (GLM) procedure..” Did you not have any random effects? I suppose only diet as fixed effect would be enough for a CRD.

Authors: we amended the statistical analysis according to the second reviewer.

Table 1- It can’t have the two diets and the supplement under the title “Diet”. What is the dietary composition of the actual diets? I can’t find it on the Tables.

Authors: Thanks for the comment. The table was corrected accordingly. The ingredients were reported in the footnotes.

Table 1- where are the ingredients (according to the Table title)?

Authors: The ingredients were reported in the footnotes.

Table 2- remove “T0 values were included as a 160 covariate. .”

Authors: we corrected it.

Table 3- how did you obtain a SEM if the number of dogs in each treatment was not the same? If you had 27 dogs, you couldn’t have had 7 dogs per treatment. SEM is only the same for all diets if you have the same number of replicates.

Authors: we agree with the reviewer; however, the number of dogs was not correct. Dogs were 28;

L243- chicken meal?

Authors: we corrected it.

It needs more discussion. What is the effect these dietary changes might have on the animals? You need to consider that values were within the reference range, and that some changes, although statistically significant, do not mean they were biologically significant. Also, a change of 0.06 vs 0.04 g/100g is not meaningful, it might as well be analytical variance.

Authors: thank you for the suggestion. We implemented the discussion.

Reviewer 2 Report

Overall the study was well conducted and will attract a lot of readers. My main concerns are the limited details in the Methods section and how the data were analyzed as described below.

Page 1

The abstract reads well.

Suggest changing title to  “Antioxidant supplementation of old dog diets …”

Page 2

Line 46 typo “could influence the health status of the animals in different ways.”

In the Introduction section, please list the antioxidants and describe the rationale of including these specific antioxidants in the diet. Since they were used previously, please briefly describe them and refer to reference 17.

In the Feed sample analysis section, please describe the dosage of the antioxidants used in this trial. I was not able to identify this information in reference 17. This is important information that will serve as a reference for future studies.

Please list the distribution of age, sex, body weight, altered status, body condition score, breed, and health status in more detail for each of the 4 groups to assure evenly distributed baseline characteristics. Please describe the living environment and the feeding protocol in more detail. Where was the study conducted? What was the feeding habit and when were the samples collected relative to the feeding time? How long were the feeding and the study? Any adverse effects or health concerns?

Line 60-65 please identify the brand and the formula of the commercial diets. We need this information to better understand the composition of the base diet without the AOX supplementation, to evaluate the impact of AOX on top of the existing nutrients and AOX in the base diets.

Line 76 please write out the full names of NDF, ADF, and ADL

Line 81 “dog daily requirements for adult maintenance.” Also, did both chicken and fish diets have exactly the same amounts of nutrient composition? Based on Table 1, they were different and thus should be listed separately. If guaranteed analysis is available for these commercial diets, please provide the source here as well.

Section 2.4 please provide the references for the analytical methods here.

Page 4

Table 1. Instead of the 3 columns, please report 4 columns of FH, CH, FH+AOX, and CH+AOX which better reflect the actual intervention, since it is unclear how much AOX was added to the diet. Please also explain what DM, NDF, ADF, ADL, CP are in the table footnote.

Page 5

Table 2. Under p-value please change 2 columns (the Cov and Diet columns) to 3 columns (diet, AOX, and the diet x AOX interaction term). Please perform a two-way ANOVA instead of one-way ANOVA to evaluate the effect of diet alone, the effect of AOX alone, and the interaction between diet x AOX, to see if the addition of AOX to two different diets differentially affect the biochemical parameters. Adding AOX to FH may have a different effect from adding AOX to CH, based on certain results such as TRI and UR. Please revise the method description in the Methods section.

Page 6

Table 3. Same comments as Table 2, please perform two-way ANOVA and report the p values for diet, AOX, and the interaction term.

Page 7

Conclusion - please perform the two-way ANOVA as suggested and report the updated results here. Be specific to the parameters affected by diet, affected by AOX, and differentially affected by the interaction between diet and AOX. Based on the new results, the inclusion of AOX may or may not have a positive effect on blood FAs depending on the protein (FH or CH).

Additional comments

A couple human studies were used as references, even though it is generally known that high fat diets elicit different responses between humans and dogs, especially in terms of cardiovascular health. Reference #35 using flavonoid-rich cocoa in humans was also discussed, even though theobromine-containing cocoa is generally considered toxic in dogs. If possible, please use a different reference specific to AOX used in this study and limit the scope of references and discussion to dogs.

Some sentences still read a little awkwardly. I suggest having a native English speaker take a look at the draft and revise it one more time before the next submission.

Author Response

Overall the study was well conducted and will attract a lot of readers. My main concerns are the limited details in the Methods section and how the data were analyzed as described below.

Page 1

The abstract reads well.

Suggest changing title to  “Antioxidant supplementation of old dog diets …”

Authors: we changed the title according to the reviewer’s suggestion

Page 2

Line 46 typo “could influence the health status of the animals in different ways.”

Authors: we corrected it.

In the Introduction section, please list the antioxidants and describe the rationale of including these specific antioxidants in the diet. Since they were used previously, please briefly describe them and refer to reference 17.

Authors: Thank you for the suggestion, we added some description in the introduction section and below the table 2.

In the Feed sample analysis section, please describe the dosage of the antioxidants used in this trial. I was not able to identify this information in reference 17. This is important information that will serve as a reference for future studies.

 Authors:  According to the reviewer’s suggestion we provided the details concerning the dosage of the antioxidants in the diets

Please list the distribution of age, sex, body weight, altered status, body condition score, breed, and health status in more detail for each of the 4 groups to assure evenly distributed baseline characteristics. Please describe the living environment and the feeding protocol in more detail. Where was the study conducted? What was the feeding habit and when were the samples collected relative to the feeding time? How long were the feeding and the study? Any adverse effects or health concerns?

 Authors: Thank you for the suggestion, we added all details in section 2.1.

Line 60-65 please identify the brand and the formula of the commercial diets. We need this information to better understand the composition of the base diet without the AOX supplementation, to evaluate the impact of AOX on top of the existing nutrients and AOX in the base diets.

 Authors: Since this information cannot be stated in the manuscript for “advertisement rules” we can only provide you the website link of the diets’ manufacturer (https://eng.forza10.com/)

Line 76 please write out the full names of NDF, ADF, and ADL

Authors: we corrected them

Line 81 “dog daily requirements for adult maintenance.” Also, did both chicken and fish diets have exactly the same amounts of nutrient composition? Based on Table 1, they were different and thus should be listed separately. If guaranteed analysis is available for these commercial diets, please provide the source here as well.

Authors: Thanks for the comment. The sentence was corrected. The right nutrient composition of the two diets is reported in the table 2

Section 2.4 please provide the references for the analytical methods here.

Authors: As far as concerns the references for the analytical methods, please consider those from 19 to 23.

Page 4

Table 1. Instead of the 3 columns, please report 4 columns of FH, CH, FH+AOX, and CH+AOX which better reflect the actual intervention, since it is unclear how much AOX was added to the diet. Please also explain what DM, NDF, ADF, ADL, CP are in the table footnote.

Authors: the table was modified in order to make it easier to read; however, we prefer to keep three columns, describing the two basal diets and antioxidant supplement used.

Page 5

Table 2. Under p-value please change 2 columns (the Cov and Diet columns) to 3 columns (diet, AOX, and the diet x AOX interaction term). Please perform a two-way ANOVA instead of one-way ANOVA to evaluate the effect of diet alone, the effect of AOX alone, and the interaction between diet x AOX, to see if the addition of AOX to two different diets differentially affect the biochemical parameters. Adding AOX to FH may have a different effect from adding AOX to CH, based on certain results such as TRI and UR. Please revise the method description in the Methods section.

Authors: thank you for the suggestion. We use a new statistical approach according to the suggestion of the reviewer. The results and discussion were modified accordingly.

Page 6

Table 3. Same comments as Table 2, please perform two-way ANOVA and report the p values for diet, AOX, and the interaction term.

Authors: see above

Page 7

Conclusion - please perform the two-way ANOVA as suggested and report the updated results here. Be specific to the parameters affected by diet, affected by AOX, and differentially affected by the interaction between diet and AOX. Based on the new results, the inclusion of AOX may or may not have a positive effect on blood FAs depending on the protein (FH or CH).

Authors: we corrected it

Additional comments

A couple human studies were used as references, even though it is generally known that high fat diets elicit different responses between humans and dogs, especially in terms of cardiovascular health. Reference #35 using flavonoid-rich cocoa in humans was also discussed, even though theobromine-containing cocoa is generally considered toxic in dogs. If possible, please use a different reference specific to AOX used in this study and limit the scope of references and discussion to dogs.

Authors: thank you for the suggestion. We implemented the discussion section.

Some sentences still read a little awkwardly. I suggest having a native English speaker take a look at the draft and revise it one more time before the next submission.

Authors: we strongly revised the English of the manuscript